# Deep scanning lysine metabolism in *Escherichia coli*

Marcelo C Bassalo[1], Andrew D Garst[2], Alaksh Choudhury[3], William C Grau[4], Eun J Oh[3], Eileen Spindler[2], Tanya Lipscomb[2] & Ryan T Gill[2,3,*] (iD)

## Abstract

Our limited ability to predict genotype–phenotype relationships has called for strategies that allow testing of thousands of hypotheses in parallel. Deep scanning mutagenesis has been successfully implemented to map genotype–phenotype relationships at a single-protein scale, allowing scientists to elucidate properties that are difficult to predict. However, most phenotypes are dictated by several proteins that are interconnected through complex and robust regulatory and metabolic networks. These sophisticated networks hinder our understanding of the phenotype of interest and limit our capabilities to rewire cellular functions. Here, we leveraged CRISPR-EnAbled Trackable genome Engineering to attempt a parallel and high-resolution interrogation of complex networks, deep scanning multiple proteins associated with lysine metabolism in *Escherichia coli*. We designed over 16,000 mutations to perturb this pathway and mapped their contribution toward resistance to an amino acid analog. By doing so, we identified different routes that can alter pathway function and flux, uncovering mechanisms that would be difficult to rationally design. This approach sets a framework for forward investigation of complex multigenic phenotypes.

**Keywords** CRISPR-Cas9; genotype–phenotype; lysine; mapping; mutagenesis
**Subject Categories** Methods & Resources; Synthetic Biology & Biotechnology
**Mol Syst Biol. (2018) 14: e8371**

## Introduction

Evolution has selected for efficient and robust metabolic and regulatory networks that prevent unnecessary metabolite biosynthesis and optimally distribute resources to maximize overall cellular fitness. The complexity of such networks, coupled with limited approaches to understand their structure and function, has broadly limited capabilities for understanding and rewiring cellular networks across a range of applications (Martin *et al*, 2003; Temme *et al*, 2012; Nielsen & Keasling, 2016). Network and pathway engineering strategies have relied primarily upon coarse approaches for modulating function (e.g., promoter swaps or complete gene knockouts) at a limited number of loci. Alternatively, adaptive laboratory evolution (ALE) approaches are often employed to produce more refined adjustments (e.g., SNPs) for manipulating pathway flux. However, ALE also leads to a larger number of unintended passenger mutations and limited mechanistic understanding of the improved phenotype (Lee & Kim, 2015). Moreover, both strategies massively under sample the combinatorial space of interest. As such, network and pathway engineering would benefit from improved approaches capable of generating a broad range of targeted mutations that can be mapped with high resolution to the pathway–network-level function, mirroring deep scanning mutagenesis strategies that have revolutionized protein engineering (Fowler & Fields, 2014; Butterfield *et al*, 2017; Chevalier *et al*, 2017; Rocklin *et al*, 2017). This capability would provide for entirely new paradigms to study and engineer complex multigenic phenotypes, exploring sophisticated hypotheses to optimize function through transcription, translation, stability, and kinetics among others that encompass the breadth of what is found in nature. Here, we take a step toward this capability by demonstrating sequence-to-function mapping at a pathway scale.

Amino acid metabolism is fundamental to all domains of life, consisting of highly evolved pathways with extensive kinetic and regulatory features, making them an ideal model system for our demonstration studies (Fig 1A). Additionally, amino acids comprise large industrial product markets—lysine, for example, is used in the animal feedstock, pharmaceutical, and cosmetics industries, comprising a multibillion-dollar market (Yokota & Ikeda, 2017). Lysine overproducers were traditionally identified via adaptation in the presence of antimetabolites such as the analog S-(2-aminoethyl)-L-cysteine (AEC). Derepression of lysine biosynthesis has been previously implicated as a mechanism of resistance to AEC (Blount & Breaker, 2006; Blount *et al*, 2007); however, the complexity of this phenotype has also implicated other mechanisms such as improper discrimination by the lysyl-tRNA synthetase machinery (Ataide *et al*, 2007). Ultimately, the underlying genetic basis of lysine overproduction and its relationship to deregulation and antimetabolite resistance provides a challenging system for genetic study. As an example, sequencing of a lysine-overproducing industrial strain of *Corynebacterium glutamicum* revealed that more than 1,000 mutations have accumulated in the

---

1   Department of Molecular, Cellular and Developmental Biology, University of Colorado Boulder, Boulder, CO, USA
2   Inscripta, Inc., Boulder, CO, USA
3   Department of Chemical and Biological Engineering, University of Colorado Boulder, Boulder, CO, USA
4   Department of Chemistry and Biochemistry, University of Colorado Boulder, Boulder, CO, USA
    *Corresponding author. Tel: +1 303 492 2627; E-mail: rygi0567@colorado.edu

genome after decades of adaptive evolution (Yang & Yang, 2017; Yokota & Ikeda, 2017). Although recent system-based approaches (Koffas & Stephanopoulos, 2005; Becker *et al*, 2011; Lee & Kim, 2015) are being used to elucidate the biochemical and regulatory mechanisms of lysine overproduction, current strategies rely on individually constructing and testing single sequence-to-activity hypotheses, requiring substantial investment in time and resources.

A powerful tool to overcome our limited ability to predict the phenotypic consequences of mutations in single proteins is to introduce every possible mutation and couple that to a genotype–phenotype assay platform, such as in the case of deep scanning mutagenesis (Fowler & Fields, 2014). As an example, Sarkisyan and collaborators (Sarkisyan *et al*, 2016) investigated tens of thousands of single and multiple mutations in the coding sequence of GFP to report a local fitness landscape for this protein. Saturation mutagenesis has also been employed in a variety of different contexts to address a range of biological and engineering questions (Findlay

*et al*, 2014; Canver *et al*, 2015; Jeschek *et al*, 2016; Chevalier *et al*, 2017). Expanding this concept to a repertoire of proteins connected to one another through a phenotype of interest would allow the parallel investigation of pathways and networks on a system scale. This requires, however, the ability to individually measure genotype–phenotype relationships for each of the designed mutants across all targeted proteins. We recently reported a method (CRISPR-EnAbled Trackable genome Engineering or CREATE) (Garst *et al*, 2017) that allows parallel mapping of mutations in a massively multiplex scale. CREATE leverages array-based oligo technologies to synthesize and clone hundreds of thousands of cassettes containing a genome-targeting gRNA covalently linked to a dsDNA repair cassette encoding a designed mutation. After CRISPR/Cas9 genome editing, the frequency of each designed mutant can be tracked by high-throughput sequencing using the CREATE plasmid as a barcode. We envisioned that with this technology, all proteins associated with a metabolic pathway could be interrogated in parallel at

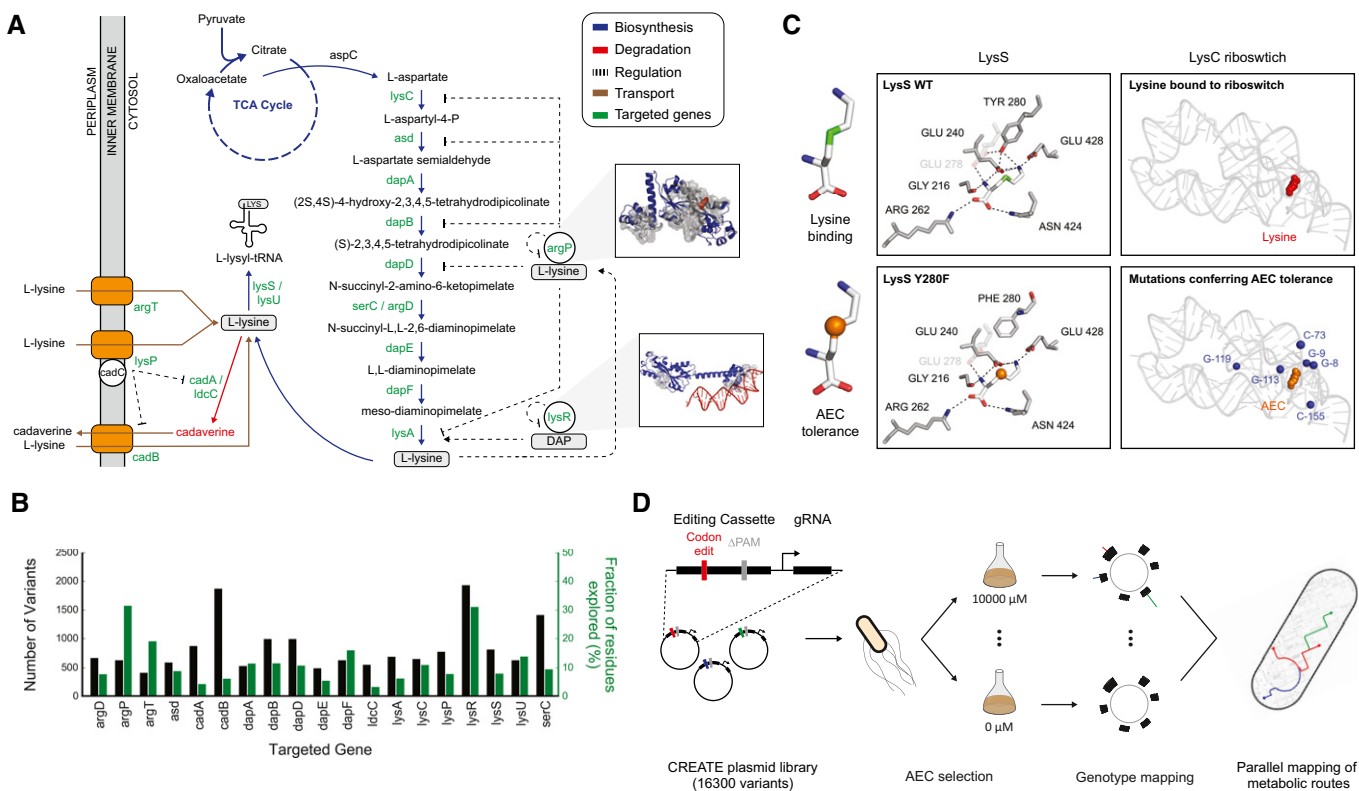

**Figure 1.  Library design and selection strategy.**

A   Overview of the lysine metabolism in *E. coli*. The arrows are color coded according to the different metabolic categories, as defined in the figure legend. Genes targeted in the library are highlighted in green. The insets represent examples of library designs for two targeted proteins, with the targeted residues included inside the gray surface representation.

B   For each targeted gene, the number of variants (black bars, left *y*-axis) and the fraction of the single substitution sequence space (green bars, right *y*-axis) are plotted. The total library size across all genes sums to 16,300 variants.

C   Description of the two main mechanisms of AEC toxicity. The structural differences between canonical lysine and AEC are shown in the left, with the orange sphere highlighting the sulfur group present in AEC. Lysine binding is shown in the top panels, and AEC is shown in the bottom panels. Mutations described to confer AEC resistance are highlighted in the bottom panels.

D   Workflow of the strategy to map trajectories of AEC resistance using CREATE. Briefly, designed cassettes were cloned, miniprepped, and transformed into strains expressing Cas9 and the lambda red machinery. The library culture was grown for 8 h in LB media with proper antibiotics, washed with PBS, and inoculated into M9 minimal media containing the AEC selective pressure and antibiotics. An aliquot was stored for initial plasmid barcode sequencing counts. After growth, cells were harvested for deep sequencing of the plasmid barcodes, which were used to map the enrichment scores of the designed mutants.

single nucleotide resolution, thus demonstrating deep scanning mutagenesis at the pathway scale.

Here, we specifically investigate lysine metabolism in *Escherichia coli*. We constructed a saturation mutagenesis library in binding pockets of key proteins involved in four main categories that affect lysine homeostasis: (i) biosynthesis, (ii) degradation, (iii) regulation, and (iv) transport (Fig 1A). By challenging this library with the antimetabolite AEC, we hypothesized that we could evaluate in parallel the contribution of these 16,300 targeted mutations toward antimetabolite resistance and thus overall pathway flux. In testing his hypothesis, we demonstrated the ability to identify mutations beyond dominant selection winners and to uncover mechanisms for altering pathway flux that would have been difficult to predict *a priori*. We also identified important factors that must be taken into consideration when attempting genotype–phenotype mapping at a pathway scale. As such, this work provides a framework for directed engineering of complex multigenic phenotypes.

## Results

### Lysine library design and selection strategy

We designed 16,300 mutations targeting four primary routes that affect lysine flux: lysine biosynthesis (12 genes), lysine degradation (two decarboxylation genes), lysine transport (three genes), and regulation of genes in such pathways (two genes; Fig 1A). For each targeted gene, we designed and constructed full saturation mutagenesis libraries of all residues within a 6 Å shell from known or model-predicted binding sites, encompassing substrate, co-factor, DNA binding, or allosteric factors. A comprehensive description of all targeted sites and the respective cassette sequences are listed in Dataset EV1. This strategy allowed us to scan probable targets for proteins with no known functional sites, and an average higher than 50% of known functional sites in the remaining proteins (Dataset EV1). Overall, 3.5–32% of all residues for each of the 19 genes

involved in lysine metabolism were fully saturated (Fig 1B). The constructed plasmid libraries were deep sequenced to confirm coverage. We observed that 99% of the designs were cloned successfully into the plasmid backbone, with 91–93% surviving after exposure to Cas9 across two biological replicates (Fig EV1).

In order to assess coverage at the genomic level and confirm that edits are indeed introduced in the genome, we deep sequenced one targeted genomic window from each of five genes across biological replicates. Overall, we measured 22.6–61.6% of the designed edits in these regions (Fig EV2, Table 1). Further calculation suggests that the overall genomic editing efficiency can be estimated at 1.6–3.7%, taking in consideration the ratio of edited reads to wild-type, as well as the probability of cells being edited at that specific locus versus the other targeted loci (Dataset EV4). These results demonstrate that we are effectively introducing edits at the targeted genomic loci.

To map mutations to lysine pathway function, we exposed this library to the lysine analog S-(2-aminoethyl)-L-cysteine or AEC. This analog competes with canonical lysine for binding to the lysyl-tRNA synthetase (LysRS; Ataide *et al*, 2007), leading to protein misfolding and reduced growth. Additionally, AEC blocks lysine biosynthesis by interacting with riboswitches, inhibiting bacterial growth in the absence of an external lysine source (Blount & Breaker, 2006; Blount *et al*, 2007; Fig 1C). We reasoned that designer mutations that influence lysine regulation and overproduction would allow lysine to outcompete AEC and thereby restore cell growth. Sequencing of the plasmid cassettes (herein referenced as barcodes) before and after growth in the presence of AEC allows parallel tracking of each designed mutant in the library, allowing us to perform highly parallel mapping of their contribution to tolerance and by inference to lysine flux (Fig 1D).

### Mapping the impact of each pathway category on tolerance and function

The lysine deep scanning mutagenesis library exhibited enhanced growth when compared to wild-type cells transformed with either a

**Table 1. Deep sequencing of selected genomic regions to confirm editing.**

| Gene | Replicate | Total edits designed | Number of edits observed | Fraction covered (%)[a] | Editing efficiency (%)[b] |
|---|---|---|---|---|---|
| lysP | I | 260 | 59 | 22.7 | 1.9 |
| lysP | II | 260 | 65 | 25.0 | 1.6 |
| lysC | I | 380 | 132 | 34.7 | 2.7 |
| lysC | II | 380 | 161 | 42.4 | 1.8 |
| dapF | I | 320 | 103 | 32.2 | 2.2 |
| dapF | II | 320 | 130 | 40.6 | 2.0 |
| lysR | I | 820 | 358 | 43.7 | 2.8 |
| lysR | II | 820 | 433 | 52.8 | 3.7 |
| argP | I | 560 | 265 | 47.3 | 2.9 |
| argP | II | 560 | 345 | 61.6 | 2.7 |

[a]"Fraction covered" is calculated by dividing the "Number of edits observed" by the "Total edits designed".
[b]Editing efficiency is an estimation that takes in consideration the fraction of the total library represented by the sequenced region, according to the equation $Eff = (edits/total)/(\omega/16,300)$, with "Eff" being the estimated editing efficiency, "edits" being the number of sequencing reads that mapped to a genomic edit in the targeted window, "total" being the total number of sequencing reads, and "$\omega$" being the "Total edits designed". The full list of all these values can be found in Dataset EV4.

non-targeting gRNA or a gRNA targeting the unrelated loci *galK* (double-stranded break control or DSB) across a range of AEC concentrations (Fig 2A). There were no significant growth differences between the non-target and DSB controls under AEC selection, suggesting that the improved growth phenotype observed in the library is not a consequence of DSB-induced adaptation (Shee *et al*, 2011). After 30 h, both negative controls began to grow in up to 1,000 μM AEC, suggesting that spontaneous mutations can also confer AEC tolerance.

After sequencing the lysine library barcodes before and after selection, the fitness contribution of each designer mutation to AEC resistance can be inferred in parallel (Fig 2B, Dataset EV2) and then summarized at the gene level. Mutations in several genes demonstrate consistent enrichment across several selective conditions (e.g., *lysP* and *dapF*). The majority of genes, however, demonstrate concentration-dependent enrichment, consistent with the expectation that different genes will affect network function to differing levels. Mutations in *dapB, lysA, and lysU* were not significantly

enriched in any of the selections performed. Note that when grown in the absence of AEC, the library has an enrichment score centered around 0 (Appendix Fig S1), indicating that growth in minimal media is not strongly biasing the library. However, the longer left tail toward negative enrichment scores suggests that some mutations in this pathway are likely deleterious.

Gene summaries were then mapped to the four design categories described earlier, resulting in a comprehensive map of trajectories leading to AEC resistance (Fig 2C and D). Mutations in transporters are the most effective resistance route, which is not surprising as any loss-of-function mutation could prevent cellular uptake of AEC from the media. The use of barcodes for each mutant enabled us to characterize beyond the dominant selection winner, uncovering the contribution of the remaining categories, as will be discussed below. This analysis provides a comprehensive map of the various strategies typically pursued in pathway–network engineering, highlighting what pathway features need to be optimized and which specific mutations could lead to

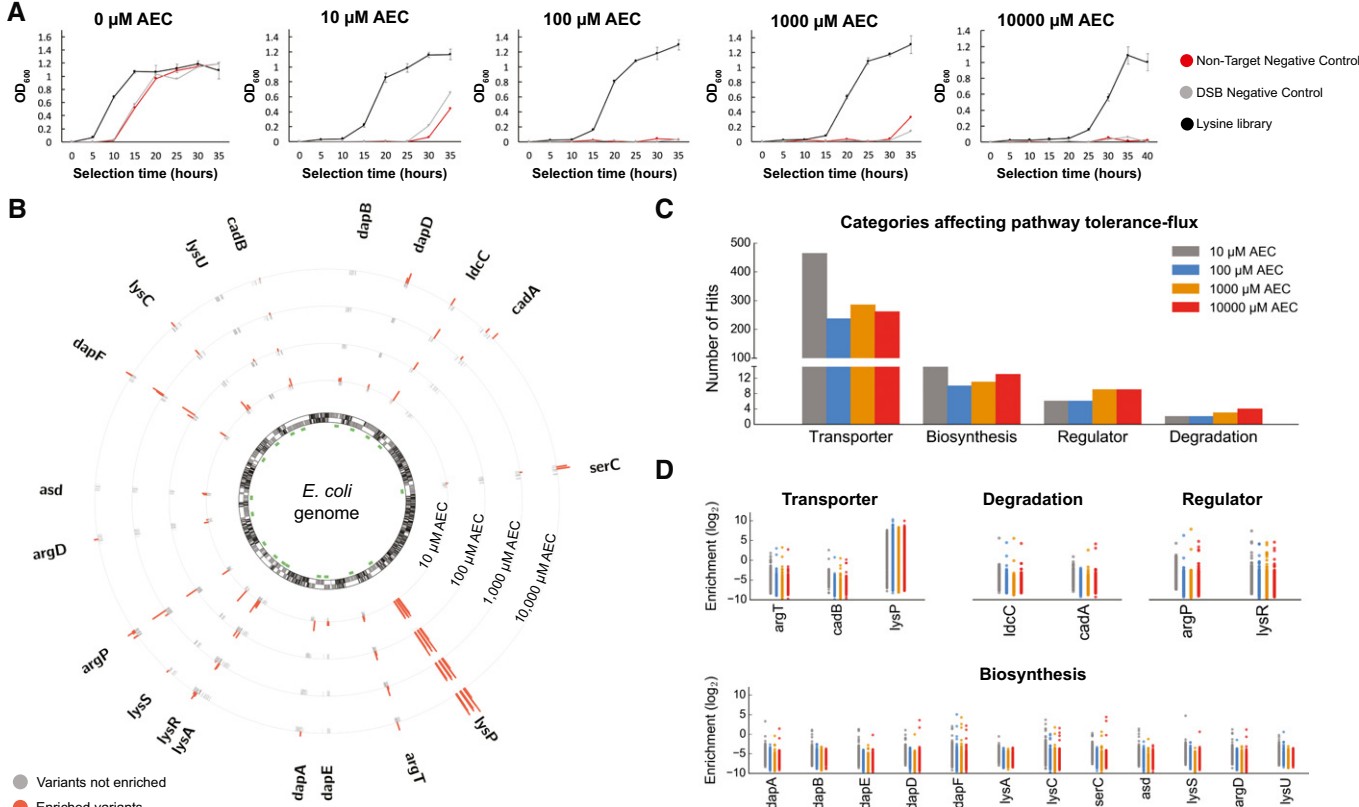

**Figure 2. Mapping the effect of each category to the lysine pathway tolerance-flux.**

A  Growth curves of the library (black) compared to two different controls under increasing selective pressures. DSB (double-stranded break) negative control is a cassette designed to introduce a stop codon at the unrelated gene *galK*. *n* = 3 for each curve. Error bars show mean value ± SD.

B  Plasmid barcode-based mapping of enriched variants across all targeted genes under AEC selection. The innermost circle represents the ORFs in the *E. coli* genome, with the green bars highlighting regions that were zoomed 50×. Positive log₂ enrichment scores of variants under increasing selective pressures are plotted as orange bars facing outward. Not enriched variants are plotted in gray bars facing inward. Two distinct biological replicates are combined in this plot, using a weighted enrichment score (described in the methods section).

C  Mapping the number of enriched mutations in genes that were classified under the different categories. The classification of each gene is the same as shown in Fig 1A.

D  Log₂ enrichment scores for each gene under each category.

phenotypic improvement. We emphasize that this map is inferred from the plasmid barcodes, which can lead to a rate of false positives as discussed in later sections. Therefore, although this approach can provide powerful insights and narrow the search space to specific targets, genomic reconstruction and validation are essential in order to be certain of the phenotypic improvement. Below, we focus on different aspects and mutations of this map, highlighting important features and limitations that need to be taken into consideration when attempting genotype–phenotype mapping at such scale.

### Transporter loss-of-function dominates the selected population

Lysine uptake is mediated by three different transporter systems in *E. coli* (Fig 1A). ArgT codes for a periplasmic binding protein specific to lysine, arginine, and ornithine, interacting with the ABC transporter coded by the *hisJQMP* operon (Nikaido & Ames, 1992). CadB is part of the Cad system, which plays a role in pH homeostasis under acidic conditions. This transporter imports lysine and excretes the decarboxylated product cadaverine in conditions of low external pH and presence of exogenous lysine (Soksawatmaekhin *et al*, 2004). Finally, LysP is a specific transporter for lysine, but also has a regulatory role in activating the Cad system through transmembrane interactions with CadC (Steffes *et al*, 1992; Tetsch *et al*, 2008). Mutations in *lysP* were identified as the most highly enriched, comprising the dominant selection winner (Fig 2B–D, Appendix Fig S2). No enrichment for *lysP* mutations was observed when cells were grown in the absence of AEC (Appendix Fig S2). This is in

accordance with previous findings that identified *lysP* mutations in AEC-resistant strains (Steffes *et al*, 1992).

When mapped at single amino acid resolution, we observed significantly enriched mutations across all targeted regions in *lysP* (Fig 3A). The relatively even distribution of enriched mutations across all targeted positions in the gene suggests loss-of-function and thereby abrogated AEC transport. These mutations map across a substantial spatial fraction of the modeled structure (Fig 3D), further supporting our speculation that they disrupt LysP function. We individually reconstructed genome-modified strains for two highly enriched mutations, T33F and Q219I. These two mutants grow similarly to wild-type cells (transformed with a non-targeting gRNA) in the absence of AEC, but exhibit superior growth under increasing AEC concentrations (Fig 3B and C).

Notably, we also observed enrichment of synonymous mutations in *lysP* under AEC selection. It is well established that synonymous mutations can have an effect on the levels, stability, and folding of both mRNA and proteins (Kudla *et al*, 2009; Hunt *et al*, 2014). As such, some synonymous mutations might alter expression or stability of LysP and thereby confer AEC tolerance. Several synonymous mutations were enriched under weak selective pressure (10 μM AEC), suggesting that small fluctuations in LysP levels may be sufficient to confer low levels of resistance (Fig 3E). As selective pressure is increased up to 10,000 μM AEC, fewer synonymous mutations were still enriched, suggesting that these mutations are introducing more drastic effects on LysP levels. Overall, the frequency of synonymous mutations affecting LysP activity is substantially higher than that observed for other targeted proteins

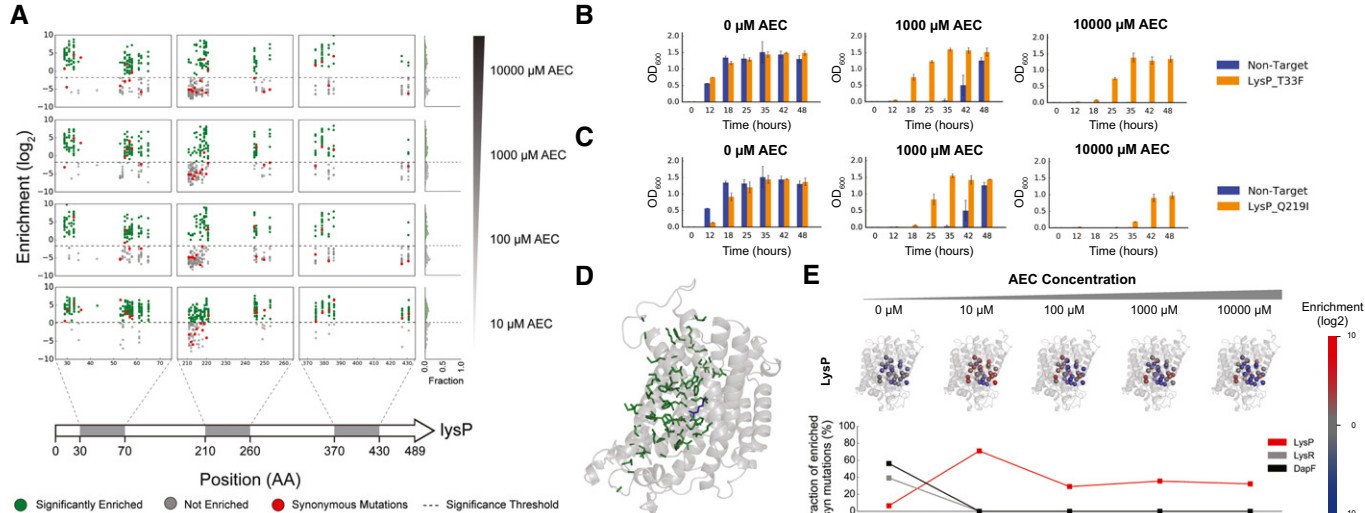

**Figure 3.  Transporter route of AEC resistance.**

A   Mapping of enrichment across the positions targeted in *lysP*. The gray regions in the gene cartoon at the bottom highlight the windows containing targeted residues, with the enrichment map shown above for increasing AEC concentrations. Enrichments are color coded according to the legend at the bottom. A histogram plot of enrichment is shown at the right.

B   Growth  of the reconstructed LysP T33F mutant compared to wild-type cells transformed with a non-target gRNA. *n* = 3. Error bars show mean value ± SD.

C   Growth of the reconstructed LysP Q219I mutant compared to wild-type cells transformed with a non-target gRNA. *n* = 3. Error bars show mean value ± SD.

D   Map of enriched mutations (green) to the modeled structure of LysP.

E   Enrichment of synonymous mutations observed for LysP. Each synonymous mutation site is shown as a sphere in the structure and is color coded according to the enrichment score bar shown on the right. The bottom chart represents the fraction of all synonymous mutations in the gene that displayed positive enrichment scores under each AEC concentration tested. For comparison, data on LysR and DapF are also shown.

(Fig 3E), highlighting an unusually strong effect of synonymous substitutions on this transporter. Since this effect is not restricted to the beginning of the gene (commonly associated with regulation of translation initiation), this result could indicate that co-translational folding is essential for LysP function. That way, changes in codon usage or disruption of important transcript secondary structures would alter ribosome attenuation sites required for proper folding (Zhang *et al*, 2009; Gorochowski *et al*, 2015). However, further studies are required to elucidate the exact mechanism.

Collectively, these results demonstrate that our deep scanning strategy maps tolerance mutations consistent with expectations (Steffes *et al*, 1992). The high fraction of *lysP* mutants in the selected population (> 95%) suggest that this is the main trajectory to evolve AEC resistance in our laboratory experiments. Directed evolution studies have demonstrated that the vast majority of mutations within a protein are known to negatively affect protein function and stability (ca. 30–50% are strongly deleterious, and 50–70% are slightly deleterious or neutral), with only a handful (0.01–1%) typically improving-altering function (Guo *et al*, 2004; Romero & Arnold, 2009; Barrick & Lenski, 2013). As such, it is not surprising that the dominant clones in our selections involved loss-of-function mutations. More importantly, this outcome highlights the importance of the use of barcoding or another method for deeply scanning selected libraries to identify a plurality of mechanisms for altering the phenotype of interest (e.g., increased pathway flux vs. decreased inhibitor flux), allowing exploration beyond a local optimum in the fitness landscape.

### Beyond the dominant selection winner: a non-obvious mechanism in DapF

With the strong dominance of *lysP* mutations (> 95%) in the selected population, identifying hits beyond the main selection winner would be challenging with traditional approaches. To demonstrate parallel tracking in this technology, we set out to validate hits in the remainder (< 5%) of the population. Among the biosynthetic genes, mutations in *dapF* were highly enriched across multiple selective pressures. This gene encodes an epimerase catalyzing the penultimate step in the biosynthetic pathway, a conversion of LL-diaminopimelate (LL-DAP) to *meso*-diaminopimelate (*meso*-DAP). DapF mutations were ranked as the most enriched non-*lysP* mutant under 100 μM AEC and the second most under 1,000 μM AEC, although no strong enrichment was observed under 10,000 μM AEC. We selected two highly enriched mutants, G210D and M260Y, for further analysis (Fig 4A).

Both G210D and M260Y substitutions lie close to the protein catalytic site (Fig 4B and C), suggesting an effect on catalytic activity. After genomic reconstruction, both mutants grew similarly to wild-type cells in the absence of AEC, but displayed distinct phenotypes when put under selective pressure. DapF G210D mutants had high growth rates up to 10,000 μM AEC (Fig EV3), confirming the barcode enrichment previously observed. However, DapF M260Y grew similarly to wild-type cells in the presence of AEC (Fig EV4). We independently retested the growth of the DapF G210D mutant, observing consistently the same phenotype of superior growth in the presence of AEC. In order to rule out adaptive mutations in *lysP*, we sequenced this locus after the selective growth and observed no mutations in this region. Mass spectrometry quantification revealed a significantly higher intracellular level

of lysine in both mutants compared to wild-type cells (Fig 4D), with G210D accumulating 51% more lysine and M260Y accumulating 111% more.

To further investigate the mechanism behind these *dapF* mutations, we purified wild-type and the mutant DapF variants (Appendix Fig S3) and measured their kinetics *in vitro* (Cox *et al*, 2002; Appendix Fig S4). Surprisingly, both DapF mutants are kinetically impaired relative to the wild-type variant (Fig 4E). We speculated that altered levels of the intermediates LL-DAP and *meso*-DAP could counterintuitively result in increased lysine accumulation through regulatory interactions. qPCR profiling of the entire biosynthetic pathway revealed one gene with statistically significant increase in gene expression, the diaminopimelate decarboxylase *lysA* (Fig 4F). LysA is responsible for the last enzymatic step in lysine biosynthesis, and it is known to be repressed by lysine (Ou *et al*, 2008; Marbaniang & Gowrishankar, 2011) and induced by diaminopimelic acid (Stragier *et al*, 1983b) through the regulator LysR. As such, the increased expression of *lysA* (Fig 4F) in a *dapF*-impaired background suggests that a larger pool of LL-DAP (previously observed in a *dapF* mutant background; Richaud *et al*, 1987) works as a stronger co-effector to activate *lysA* than the wild-type mixture of both LL-DAP and *meso*-DAP.

Overall, these results uncovered a counterintuitive interplay between lower kinetics and lysine overproduction. This finding highlights our limited ability to predict genotype–phenotype relationships in the context of an entire pathway, similar to what has been observed in the protein engineering field. Therefore, deep scanning mutagenesis proves to be a valuable strategy to identify novel regulatory mechanisms on pathway scale. Further studies are required in order to investigate the mechanistic basis for the differences in AEC tolerance between the G210D and M260Y substitutions. Other biosynthetic genes identified in our screen include *lysC*, *serC*, and *dapD*, but were not investigated in detail.

### Validating other hits: decoupling noise from real enrichment

Since plasmid barcodes are used as a proxy for genomic edits, lack of correlation introduces noise that can lead to false positives in the enrichment scores. In theory, plasmid-genome correlation should be strong for real hits with strong enrichment and weaker for non-enriched variants. To investigate this further, we focused now on the regulator category and investigated a weakly enriched mutation in LysR, as well as a strongly enriched mutation in ArgP.

Regulatory mutations are well known to confer AEC resistance (Blount *et al*, 2007; Marbaniang & Gowrishankar, 2011), mainly in the lysine-regulated riboswitch controlling expression of the aspartokinase *lysC* (Di Girolamo *et al*, 1988; Patte *et al*, 1998; Garst *et al*, 2008; Fig 1C). The regulator LysR, which upon binding to diaminopimelic acid activates the last enzymatic step in lysine biosynthesis (Stragier *et al*, 1983a; *lysA*, Fig 1A), exhibited few weakly enriched mutations in our library (Fig 5A). We focused on the LysR S36R substitution, a mutant that had significant enrichment scores at 1,000 μM AEC (*P*-value: 0.007), while at lower concentrations enrichment was not significant (*P*-value of 0.14 at 10 μM AEC and 0.12 at 100 μM AEC).

The LysR family of transcription regulators is ubiquitous in bacteria and comprises a conserved N-terminal helix-turn-helix

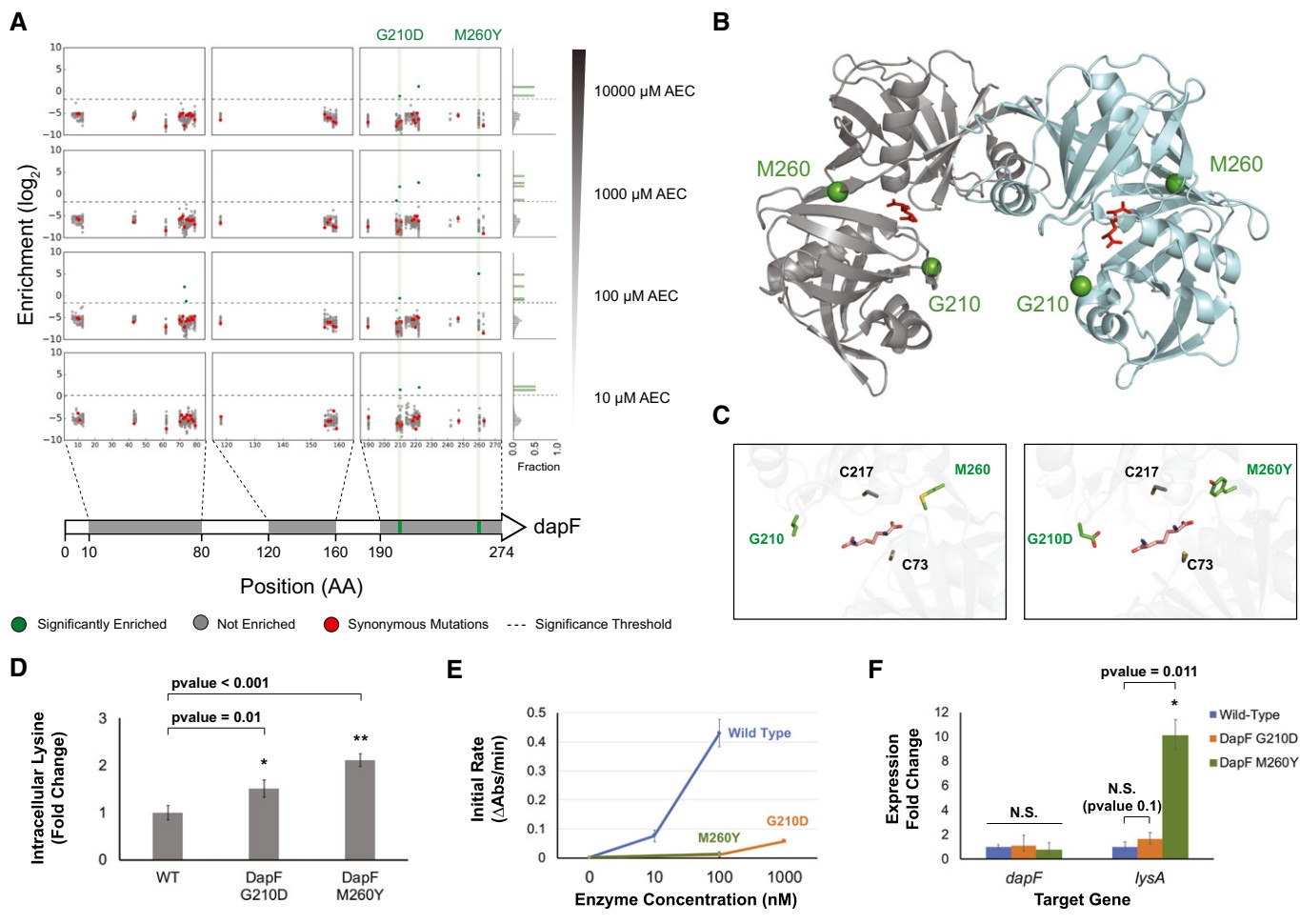

**Figure 4.  Investigating the biosynthetic gene *dapF*.**

A   Mapping of enrichment across the positions targeted in *dapF*. The gray regions in the gene cartoon at the bottom highlight the windows containing targeted residues, with the enrichment map shown above for increasing AEC concentrations. Enrichments are color coded according to the legend at the bottom. A histogram plot of enrichment is shown at the right.

B   Structure of the DapF dimer (PDB ID: 4IJZ) with the G210 and M260 sites highlighted in green. Diaminopimelate binding is shown in red.

C   Zoom in the catalytic site highlighting the G210 and M260 sites relative to the catalytic cysteines. The G210D and M260Y substitutions are shown in the right panel.

D   Absolute quantification of intracellular lysine concentration in wild-type and the reconstructed DapF mutants. Quantification was performed using LC-MS, as described in the methods section. *n* = 3. Error bars show mean value ± SD. A two-sample Student's *t*-test assuming unequal variances was performed to calculate statistical significance. Concentrations are reported as fold change relative to the wild-type control samples.

E   DapF assay showing kinetics of the wild-type, G210D and M260Y mutants. Assay was performed as described in the methods section. *n* = 5. Error bars show mean value ± SD.

F   Differential gene expression quantified via qPCR for the *dapF* and *lysA* genes on a WT, DapF G210D, and DapF M260Y backgrounds. Error bars represent 95% confidence intervals. A two-tailed Student's *t*-distribution was used to calculate *P*-values, which were adjusted using the Benjamin–Hochberg statistical method for false discovery rates.

(HTH) DNA-binding domain and a less conserved C-terminal co-inducer binding domain (Maddocks & Oyston, 2008). The LysR S36R mutation lies on the DNA-binding (HTH) domain (Fig 5B). However, after reconstruction and genomic verification of this edit, we observed that mutants do not display any alteration in intracellular lysine levels (Fig 5C). Further, we noted that strains harboring the S36R mutation grew slower than wild-type cells transformed with a non-targeting gRNA (Fig EV5). These results suggest that the enrichment observed at the plasmid barcode level for LysR is possibly a false positive.

On the other hand, the ArgP regulator displayed much stronger enrichment scores for a E246Q substitution (Fig 5D), with a *P*-value

of $1.6 \times 10^{-6}$ at 100 μM AEC, $8.1 \times 10^{-8}$ at 1,000 μM AEC, and $1.59 \times 10^{-5}$ at 10,000 μM AEC. ArgP, which also belongs to the LysR family of transcriptional regulators, can bind to lysine in order to inhibit transcription of several genes in the biosynthetic lysine pathway (Fig 1A), acting as one of the main negative feedback mechanisms (Marbaniang & Gowrishankar, 2011). The E246Q substitution lies on the C-terminal co-inducer binding domain (Fig 5E), although the apparent role for this residue is unclear. After genomic reconstruction, we observed that strains harboring the ArgP E246Q mutation accumulated 124% more intracellular lysine (Fig 5F), likely responsible for the barcode enrichment previously observed, although the reconstructed mutant could also not

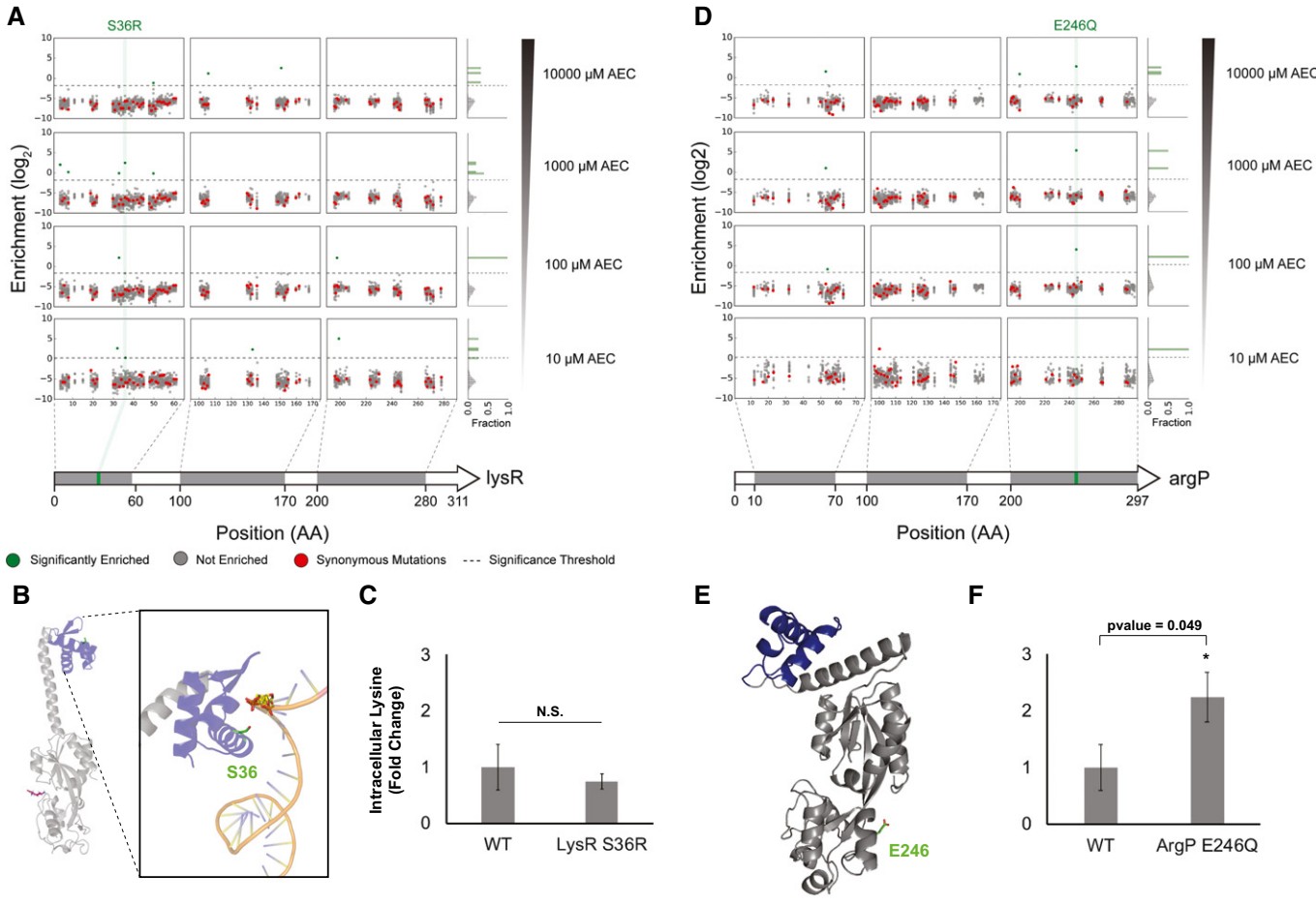

**Figure 5.  Investigating noise and real enrichment in the plasmid barcodes.**

A  Mapping of enrichment across the positions targeted in *lysR*. The gray regions in the gene cartoon at the bottom highlight the windows containing targeted residues, with the enrichment map shown above for increasing AEC concentrations. Enrichments are color coded according to the legend at the bottom. A histogram plot of enrichment is shown at the right.

B  Modeled structure of LysR, with the HTH DNA-binding domains colored in blue and the co-inducer binding domain in gray. The S36 site is highlighted in green. The right panel zooms to the S36 site, showing close proximity to the DNA phosphate backbone.

C  Absolute quantification of intracellular lysine levels in wild-type and the reconstructed LysR S36R mutant. Quantification was performed using LC-MS, as described in the methods section. $n = 2$. Error bars show mean value ± SD. A two-sample Student's *t*-test assuming unequal variances was performed to calculate statistical significance. Concentrations are reported as fold change relative to the wild-type control samples.

D  Mapping of enrichment across the positions targeted in *argP*. Representation is the same as described in (A).

E  Modeled structure of ArgP, highlighting the E246 residue in the co-inducer binding domain (gray).

F  Absolute quantification of intracellular lysine levels in wild-type and the reconstructed ArgP E246 mutant. Quantification was performed using LC-MS, as described in the methods section. $n = 2$. Error bars show mean value ± SD. A two-sample Student's *t*-test assuming unequal variances was performed to calculate statistical significance. Concentrations are reported as fold change relative to the wild-type control samples.

outcompete the wild-type strain (Fig EV6), similarly to the results observed for the DapF M260Y mutation.

In all, these results support our initial hypothesis that strongly enriched mutations are more likely to yield a real signal than mutations displaying weak enrichment scores. However, as discussed in the next section, adaptive mutations could also introduce noise in the form of strong enrichment scores. Therefore, genomic reconstruction and validation are essential in order to confirm targets identified by this approach. Further, a more stringent *P*-value threshold with improved statistical methods could filter a larger fraction of false positives in the sample. In the discussion, we highlight important practices and advances that can improve the signal-to-noise ratio in future implementations of this technology.

**Deep scanning mutagenesis provides better genotype–phenotype mapping than adaptive evolution**

The data presented herein demonstrate an ability to investigate specific sequence-to-activity hypotheses at a scale orders of magnitude beyond alternative strategies. To further justify this claim, we performed adaptive laboratory evolution and whole genome sequencing under a selective AEC concentration (1,000 μM). Specifically, we adapted wild-type *E. coli* cells and fully sequenced the genomes from 15 isolates after 2 days (single-batch) or 5 days (serial transfer) of selection. As expected, multiple SNPs (2-7 SNPs per genome post-filtering) were identified (Fig 6A, Dataset EV3). Only one gene in the lysine pathway was found to be mutated (*lysP*), with

five distinct SNPs identified in a total of eight occurrences. The remaining 21 distinct SNPs totaled 48 occurrences and were spread across a broad range of categories (Fig 6B).

Overall, these whole genome sequencing studies affirm the well-established ratio of positive to neutral to negative mutations observed in laboratory evolution. The low fraction < 0.5–1% of positive mutations increases the subsequent screening burden (all individual mutants must be reconstructed and tested) by 2 orders of magnitude (1/(0.5–1%) = 100–200×). Moreover, most (80%) of the identified mutations do not map to genes reasonably linked to the pathway (Fig 6C), thus challenging any rational strategies for reducing the reconstruction and screening burden. While only the dominant selection winner (*lysP*) was uncovered using adaptive evolution, CREATE provided much higher depth for the regions targeted in the library, effectively scanning these pre-selected hotspots (Fig 6D). We emphasize the value of a combination of such approaches: Adaptive evolution could be leveraged to evolve complex phenotypes and inform putative target genes, and CREATE could be leveraged to reconstruct the identified mutations and investigate their individual contributions in parallel.

## Discussion

Complex phenotypes are often engineered through directed evolution or other random mutagenic strategies. While successful for phenotype optimization in industrial strains, off-target mutations can decrease overall cell fitness and lead to "dead-end" phenotypes, preventing further improvement of the evolved strain (Lee & Kim, 2015). New tools (Garst *et al*, 2017; Bao *et al*, 2018; Guo *et al*, 2018; Roy *et al*, 2018; Sadhu *et al*, 2018) that combine targeted deep scanning mutagenesis with genotype–phenotype mapping provide a powerful framework to explore distinct hypotheses in parallel, uncovering mechanisms that would be difficult to rationalize in

complex systems. This concept was evident for the DapF mutations investigated here, in which lower kinetics counterintuitively improved lysine accumulation in the strains.

Further, the ability to map deeply, through the use of barcodes, enabled quantification beyond the main selection winner. Transporter loss-of-function was a clear solution to the AEC challenge, dominating most of the selected population. Therefore, looking beyond *lysP* mutations would be challenging with traditional strategies (Fig 6). However, we could correctly identify other hits that were being masked by the enrichment of *lysP* mutations, highlighting the value of parallel genotype–phenotype mapping. We note that some of the mutants described here could not outcompete adaptive mutations that inactivated *lysP*, growing similarly to wild-type cells even though a clear improvement in lysine accumulation was observed. This finding underlines the complex relationship between the selection environment and the fitness effect. In unicellular asexual organisms such as bacteria, fitness in a competitive environment can be mainly attributed to three parameters: (i) lag phase duration, (ii) exponential growth rate, and (iii) maximum yield at saturation. Mutations can affect fitness through differing degrees on each of these parameters (Gall *et al*, 2008; Adkar *et al*, 2017). Moreover, the effect of each parameter is further confounded in more complex populations, in which clonal interference has a strong effect on shaping the adaptation dynamics and evolutionary outcomes (Barrick & Lenski, 2013; Lang *et al*, 2013). These different adaptive niches could explain the results observed here, and recently developed tools could aid in the elucidation of these evolutionary niches at the population scale (Wong *et al*, 2018).

Although successful in mapping different AEC resistance routes in parallel, several false-positive mutants were identified in our studies. These could have been a consequence of background adaptive evolution, or potentially from imperfect plasmid barcode to genome-edit correlation. Additionally, a key limitation on this process is the generally low editing efficiency on a library scale

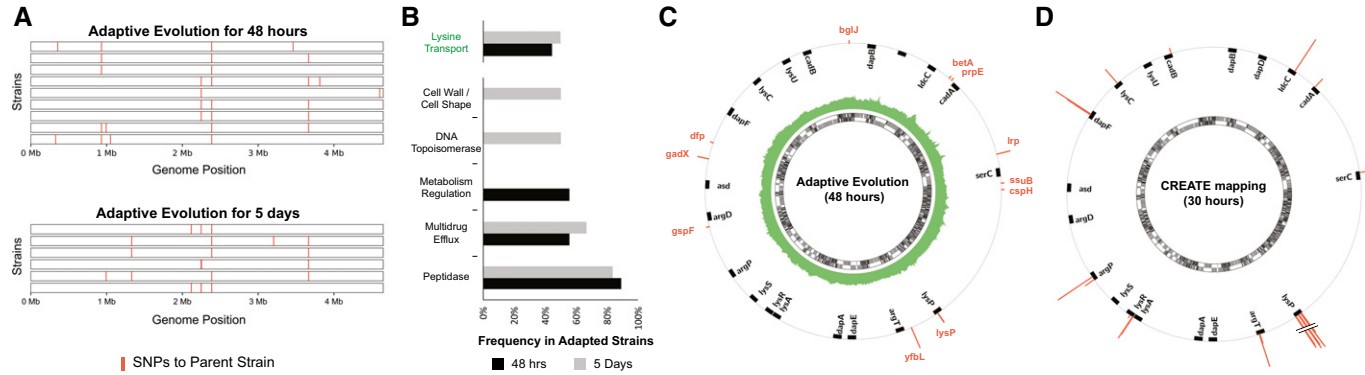

**Figure 6. Comparison of mapping depth using adaptive evolution and the deep scanning mutagenesis library.**

A  Map of SNPs positions observed in each sequenced genome after 48-h adaptation (n = 9) and 5-day adaptation with one passage per day (n = 6). SNPs were mapped to the parent strain, sequenced after growth in minimal media in the absence of AEC.
B  Categories of the SNPs found in Fig 6A, with the categories from genes that are directly linked to the lysine pathway highlighted in green.
C  Circos plot of the SNPs found after 48-hour adaptation relative to the lysine metabolism genes. The bar plots of each SNP represent the frequency across sequenced strains (n = 9). Lysine metabolism genes are highlighted as black bars (50× zoom). The inner green plot represents the average sequencing coverage for each position across all sequenced genomes.
D  Map of enriched mutations found for the same selective pressure (1,000 μM AEC) using our deep scanning mutagenesis library. The bar plots represent log₂ enrichment scores.

(Table 1), which is a consequence of multiple variables in the editing process. First, variations between gRNA activity and the ability to rescue double-stranded breaks (DSB) via homology-directed repair are major drivers of fluctuations on editing efficiency. Additionally, many different escape mechanisms can prevent proper function of the CRISPR/Cas9 machinery, such as mutations in the Cas9 or the gRNA itself. Errors in oligo synthesis can further prevent the introduction of a DSB, by incorporating mutations in the gRNA sequence for example. Finally, wild-type cells that escape the DSB process through any of the mechanisms above are inherently more fit, since they do not undergo the toxicity and stresses caused by DNA damage.

With these identified limitations, a few parameters must be taken into consideration when attempting genotype–phenotype mapping on a pathway scale. First, applications that include strong selective pressures are more likely to succeed. With the relatively low (2–4%) editing efficiencies reported herein, the screening burden would be too high for most screening throughputs. Second, while these technologies efficiently narrow the search space to a few hypotheses (genes and specific mutants) of interest, reconstruction in wild-type backgrounds and subsequent validation are essential. Third, the use of multiple biological replicates is fundamental to deconvolute designed edits from adaptive evolution background, so that barcodes displaying enrichment in different samples are more likely to be real. Fourth, sequencing depth remains an important consideration. In this study, the selective dominance of *lysP* mutations likely prohibited the investigation of every single designed edit. A rarefaction curve should be included in future studies in order to assess the required sequencing depth. Finally, strategies to improve map accuracy would be valuable additions. As an example, the use of single cell-specific barcodes could improve the confidence of mapping, so that each single mutation is mapped as a population of cells (Zeitoun *et al*, 2017). Transferring barcodes from plasmids to genomes (Roy *et al*, 2018) could also decrease cell-to-cell variation and hence decrease noise in barcode enrichments. In the specific case of lysine metabolism, comparing mutations identified in the presence of different antimetabolites or with screening-based approaches using lysine biosensors (Yang *et al*, 2013; Wang *et al*, 2015, 2016) would be a valuable contribution.

Overall, we demonstrated the expansion of deep scanning mutagenesis strategies from a single gene to an entire metabolic pathway. We identified in parallel multiple routes of AEC resistance, encompassing mutations in transporters, regulators, and biosynthetic genes. This technology, as well as future implementations that address some of the limitations described above, should accelerate our ability to investigate complex multigenic phenotypes, providing knowledge that will contribute to the forward engineering of these traits.

# Materials and Methods

### Genome-edited strains, plasmids, and general cloning procedures

Genome editing and individual mutant validation were performed in a wild-type *Escherichia coli* str. K-12 substr. MG1655 strain. A custom pSIM5-Cas9 dual vector was built by cloning the araC-pBAD-Cas9 fragment from pX2-Cas9 vector (Addgene #85811) into the temperature-sensitive pSIM5 plasmid (Datta *et al*, 2006) containing the lambda red genes. This pSIM5-Cas9 dual vector was transformed into *E. coli* MG1655 prior to the library introduction. The editing cassettes containing the homology arm and genome-targeting gRNA were cloned in the same backbone previously used for CREATE (Garst *et al*, 2017) (example vector can be visualized here: https://benchling.com/s/seq-mrFmtCypVLPiiJ4lJk3T).

Cloning procedures that did not involve libraries were performed using CPEC (Quan & Tian, 2011). Briefly, fragments containing 40bp homology arms were PCR amplified using Phusion High-Fidelity PCR Master Mix (New England Biolabs), treated with DpnI to remove methylated plasmid templates when necessary, and purified from 1% agarose gels using the QIAquick Gel Extraction Kit (QIAGEN). CPEC assembly was performed using 300 ng of backbone and equimolar insert amounts. After 10 cycles of reaction, the product was dialyzed and transformed via electroporation into E. cloni 10GF' ELITE Electrocompetent Cells (Lucigen). Cloning procedures for the library preparation will be detailed below.

### Library design

For each targeted protein in this study, 3D structures were collected from the RCSB Protein Data Bank (Berman *et al*, 2000) if available or modeled using SWISS-MODEL (Arnold *et al*, 2006) or I-TASSER (Roy *et al*, 2010). A 6 Å shell from binding sites was built using PyMOL (v.1.8.6.2) scripts to select sites for mutagenesis. A comprehensive list of all selected sites and structure details can be found in Dataset EV1. In total, 19 genes and 815 sites were selected. For each selected site, a full codon saturation mutagenesis was introduced using the most frequent codons, resulting in a total of 16,300 variants. For each variant, the gRNA and homology arm designs were automated using previously described Python scripts (Garst *et al*, 2017). Briefly, the cassette design included the following features: a library-specific 18 nt priming site for subpooling, a 12 nt variant-specific priming site (not used in this study), a 118 nt homology arm encoding the specific genomic edit and a synonymous PAM mutation in close proximity, the constitutive promoter J23119 (35 nt), a 3 bp spacing sequence (ATC), the 20 nt spacer region required for Cas9 targeting, followed by 24 nt of the 5′ end of the canonical *S. pyogenes* gRNA. The full list of cassette sequences can be found in Dataset EV1.

### Library construction

The designed library was synthesized as 230-mers by Agilent Technologies in a custom array and delivered pooled as lyophilized single-stranded DNA. As described in more details previously (Garst *et al*, 2017), the oligo pool was subjected to an Alexa Fluor 488-labeled strand extension reaction and purified in a 6% SDS–PAGE gel to remove indels introduced in the synthesis process. From the resulting purified oligo pool, the lysine library was amplified as a single subpool using predefined library-specific priming sites included in the cassette design. The amplification was optimized to minimize overamplification in an effort to reduce product crossover. The PCR was performed using Phusion High-Fidelity PCR Master Mix (New England Biolabs) and the following reaction conditions: 98°C for 60 seconds, followed by eight cycles of $98°C_{30s}/68°C_{30s}/72°C_{90s}$, followed by 10 cycles of $98°C_{30s}/72°C_{90s}$, and then a

final extension at 72°C for 3 min. The library product was purified from 1% agarose gels using the QIAquick Gel Extraction Kit (QIAGEN).

The amplified library was cloned using Gibson Assembly HiFi 1-Step Kit (SGI-DNA), with 300 ng of the linearized backbone and 30 ng of the library insert. The cloning reaction was dialyzed and then transformed via electroporation into E. cloni 10GF' ELITE Electrocompetent Cells (Lucigen), in a single electroporation using a 0.2-cm-gap cuvette (Gene Pulser, Bio-Rad). Cloning efficiency was estimated by counting colonies in LB agar plates. Overall, $> 60\times$ coverage (total CFUs/number of library variants) was achieved at the cloning stage. Subsequently, the library was grown in LB media to saturation and plasmid was extracted using the QIAprep Spin Miniprep Kit (QIAGEN). The plasmid library was then transformed into *E. coli* MG1655 following a modified recombineering protocol (Sharan *et al*, 2009). Briefly, the strain previously transformed with the dual Cas9/pSIM5 vector was grown at 30°C in LB media in 250-ml flasks under 200 rpm until mid-log phase ($OD_{600} = 0.4–0.5$). Cells were then induced with 0.2% arabinose (for Cas9 induction) and placed in a 42°C shaking water bath for 15 min (for lambda red induction). Next, cells were kept on ice for 5 min and made electrocompetent. To ensure coverage, 2 μg of the plasmid library was transformed in a single electroporation using a 0.2-cm-gap cuvette (Gene Pulser, Bio-Rad). Two independent transformations were performed for the library (biological duplicates), followed by recovery in 5 ml of LB media supplemented with 0.2% arabinose for 3 h at 30°C. Afterward, cells were plated in LB media with the proper antibiotics to calculate transformation efficiency and transferred to 30 ml of liquid LB media with antibiotics for 8 h before proceeding to selective conditions. Overall, $> 300\times$ coverage was achieved at this stage (total CFUs/number of library variants). Both the cloning and recombineered libraries were sequenced using an Illumina MiSeq run to assess the real plasmid library coverage (threshold set at 100% full matching cassettes, Fig EV1). Deep sequencing procedures for plasmid libraries are detailed below.

## AEC selections and high-throughput sequencing of the library barcodes

Selection was performed in 30 ml of M9 minimal media containing 5X M9 Minimal Salts (BD Biosciences), 2 mM magnesium sulfate, 0.1 mM calcium chloride, 1% glucose, 100 μg/ml carbenicillin (to select for the library plasmid), and varying S-(2-aminoethyl)-L-cysteine (AEC) concentrations (0–10,000 μM). The library culture growing for 8 h in LB media (described above) was washed with PBS, and 10 μl was used to inoculate the selective media. Cultures were kept at 37°C under 200 rpm. Two different selection controls were included, all subjected to the same construction procedure described above: (i) a non-targeting control, containing a plasmid with a gRNA that does not target the *E. coli* genome and (ii) a double-stranded break control, containing a plasmid with a CREATE cassette designed to introduce a stop codon at the unrelated gene *galK*.

Selections up to 1,000 μM AEC were harvested to sequence the library barcodes at 30 h post-inoculation, and the 10,000 μM AEC selections were harvested at 40 h post-inoculation. To do so, 3 ml of the selection cultures was pelleted and plasmid DNA was extracted using the QIAprep Spin Miniprep Kit (QIAGEN). Next,

custom Illumina-compatible primers (Garst *et al*, 2017) were used to barcode each selection using Phusion High-Fidelity PCR Master Mix (New England Biolabs), 300 ng of the plasmid prep, 3% DMSO, and the following cycling conditions: 98°C for 30 s, 20 cycles of $98°C_{10s}/68°C_{15s}/72°C_{20s}$, followed by a final extension of 72°C for 5 min. PCR products were purified from 1% agarose gels using the QIAquick Gel Extraction Kit (QIAGEN), pooled together in equimolar amounts, and sequenced using an Illumina MiSeq 2x150 paired-end reads run.

## Processing of the library barcode reads and statistical analysis

Reads were demultiplexed and then merged using the PANDAseq assembler (v2.10). Merged reads were matched to the database of all designed cassettes using the usearch_global algorithm (v9.2.64), with an identity threshold of 95% and minimal alignment length of 150 bp. These parameters were chosen so that chimeras in the designs could be evaluated. Forty possible hits were allowed for each query, which were subsequently sorted by percent identity, and the best-matching cassette was chosen. To generate read counts for each designed cassette, only reads that had a full alignment and an identity higher than 99% were used. The number of reads obtained at each processing step is outlined in Dataset EV4.

The next processing steps of the read counts were done using the Pandas data analysis Python package (v0.20.2). First, since low-count variants are subject to counting error, variants with initial counts (pre-selection) of less than 10 were not included in the individual biological replicate analysis. Then, variants with 0 counts post-selection were replaced to 0.5 in order to allow the subsequent calculation steps. For each individual biological replicate, enrichment scores were calculated as the logarithm (base 2) of the ratio of frequencies between post-selection and pre-selection. Frequencies were determined by dividing the read counts for each variant by the total experimental counts. Finally, a weighted average was used to combine the enrichment scores obtained in the two biological replicates, according to the formula:

$$W_{\text{avg}} = \frac{\sum_{i=1}^{N}(C_i \times W_i)}{\sum_{i=1}^{N} C_i}$$

where $W_{\text{avg}}$ is the weighted average score, $i$ is the biological replicate, $C$ is the read count obtained for the variant in the biological replicate, and $W$ is the enrichment score calculated for the variant in the biological replicate.

To assess significance, the average of enrichment scores for all synonymous mutations included in the library was calculated (average μ of wild-type enrichment). Bootstrap analysis (resampled with replacement 20,000 times) was performed to obtain a 95% confidence interval for the wild-type enrichment average μ. Variants were considered as significantly enriched if their weighted enrichment scores were at least μ ± 2*σ (i.e., *P*-value ≤ 0.05 assuming a normal distribution of synonymous mutations enrichment scores), with σ being the standard deviation. For individual mutants chosen to be investigated further in this study, the *P*-value of their respective enrichment scores was calculated using the probability density function of all mutants under the specific selective pressure.

## Deep sequencing of selected genomic regions

Genomic pockets were PCR amplified using primers annealing specifically to the target genomic region (Fig EV2). To these primers, the Nextera adapter sequences were included as 5′ overhangs, resulting in the Forward primers: 5′-TCGTCGGCAGCGTCAGATGTGTATAAGAGA CAG-[priming site]-3′ and Reverse primers: 5′-GTCTCGTGGGCTCG GAGATGTGTATAAGAGACAG-[priming site]-3′. Samples were then prepared with the Nextera XT DNA Library Prep Kit (Illumina) and sequenced on an Illumina NextSeq 2x150 paired-end reads run. Sequencing reads were merged using the PANDAseq assembler (v2.10) and trimmed to the positions highlighted in Fig EV2 (these positions exclude the primer binding site). A database was generated containing all expected sequence variants for the full length between the sequenced positions, which is the wild-type sequence and all designed edits incorporated into the respective positions. Reads were then matched at 100% identity to this database using custom Python scripts. The number of reads obtained at each processing step is outlined in Dataset EV4.

## Individual mutant reconstruction

To individually reconstruct the mutants investigated in this study, the same cassette sequence included in the library for that specific variant was ordered separately as a gblock from Eurofins Genomics. The cassette was then cloned, sequence verified, and introduced in *E. coli* MG1655 using the same procedure described above. Then, the specific genomic edit was confirmed through Sanger sequencing of the target site.

## Absolute quantification of intracellular lysine levels

Saturated overnight cultures of the reconstructed mutants were used to inoculate 100 ml of the minimal media used for selections (without any AEC present). Inoculums were made to an initial $OD_{600}$ of 0.01, and cultures were grown in shake flasks at 37°C under 200 rpm until $OD_{600}$ reached 0.5. At this stage, cells were plated to calculate CFUs/ml, washed with PBS, pelleted by centrifugation, and stored at −80°C for metabolite extraction. The frozen cell pellets were extracted in ice-cold lysis buffer, a 5:3:2 ratio of MeOH:ACN:H2O, containing amino acid standard mix at a final concentration of 1 μM (MSK-A2-1.2 standard amino acid mix, purchased from Cambridge Isotope Laboratories, Inc., Tewksbury, MA). Samples were vortexed for 30 min at 4°C with 1-mm glass beads. Insoluble proteins and lipids were pelleted by centrifugation at 4°C for 10 min at 12,000 *g*. Supernatants were collected and analyzed using a Thermo Vanquish UHPLC coupled online to a Thermo Q Exactive mass spectrometer. UHPLC-MS methods and data analysis approaches were performed as described previously (Nemkov *et al*, 2015). The intracellular concentration of wild-type control samples was normalized to 1, and the experimental samples are reported as fold change relative to these wild-type levels.

## Expression and purification of the DapF mutants

The *dapF* variants were PCR amplified from boiled cells that contained the desired mutation (wild-type *E. coli* MG1655 for the wild-type *dapF* sequence; reconstructed *dapF* mutants for the G210D and M260Y variants). The PCR products were then cloned and sequence verified into a custom-made pET-3 backbone, containing the histidine tag (6×) on either the 5′ or 3′ end of the genes to test for optimal expression. *Corynebacterium glutamicum* DAP dehydrogenase was synthesized from Eurofins Genomics and also cloned in the pET-based vector. Expression was done in a *E. coli* BL21 strain using LB media, which was induced with 1 mM IPTG when $OD_{600}$ reached 0.6. Induced cultures were grown at 30°C overnight under 200 rpm, harvested by centrifugation, and the pellet stored at −80°C for protein purification.

Proteins were purified using the Ni-NTA Spin Kit (QIAGEN), following the protocol for purification of tagged proteins under native conditions. Purified samples were run on a denaturing PAGE gel (Mini-PROTEAN TGX Stain-Free Precast Gels, Bio-Rad) to confirm purity and quantified using the Thermo Fisher Scientific Pierce 660 nm Protein Assay Reagent. Purified proteins were used fresh for the kinetic assay (never frozen).

## *In vitro* assay to measure DapF kinetics

Enzymatic activity of the DapF variants was determined *in vitro* using a modified DAP epimerase–DAP dehydrogenase coupled spectrophotometric assay (Cox *et al*, 2002). Briefly, 100 mM Tris (pH 7.8), 0.1 mM diaminopimelic acid (racemic mixture), 0.44 mM $NADP^+$, and 1 mM DTT were added to a cuvette and incubated at 37°C for 10 min to equilibrate the temperature. Then, 1.8 mM DAP dehydrogenase was added and the absorbance was recorded at 340 nm until it reached a plateau (i.e., all meso-DAP was depleted; Appendix Fig S4). Next, varying amounts of the purified DapF variants were added, and the absorbance at 340 nm followed through time. The assay was performed with 400 μl final volume in a Nano-Drop One$^C$ UV-Vis Spectrophotometer (Thermo Fisher Scientific Inc.).

## Quantitative analysis of gene expression

Wild-type *E. coli* MG1655 and the analyzed reconstructed mutants were grown under the same conditions as described for absolute intracellular lysine quantification. At the harvest stage ($OD_{600}$ = 0.5), 1 ml of the culture was treated with RNAprotect Bacteria Reagent (QIAGEN) to stabilize the RNA and the resulting pellet frozen at −80°C. Total RNA was then extracted using the RNeasy Mini Kit (QIAGEN) with an on-column DNase digestion. cDNA was synthesized using the SuperScript IV First-Strand Synthesis System (Invitrogen). Power SYBR Green Master Mix (Thermo Fisher Scientific Inc.) was then used for the qPCRs, which was run on a QuantStudio 6 Flex Real-Time PCR System (Thermo Fisher Scientific Inc.) with the following conditions: 95°C for 30 s, 40 cycles of $95°C_{30s}/65°C_{30s}/72°C_{30s}$, followed by the standard melting curve protocol. Three different housekeeping genes were tested as qPCR endogenous controls: the 5S ribosomal RNA (*rrfA*), siroheme synthase (*cysG*), and the integration host factor B (*ihfB*). After testing each endogenous control, *ihfB* exhibited variability among samples, and so *rrfA* and *cysG* were chosen as endogenous controls for the analysis. Relative expression was calculated using the $\Delta\Delta C_t$ method (Livak & Schmittgen, 2001) on the Thermo Fisher Cloud Data Analysis Apps (qPCR Module).

## Adaptive evolution and whole genome sequencing

The adaptive evolution experiments were performed with wild-type *E. coli* MG1655 (without any plasmids) in 30 ml of the same minimal media used for selections, containing 1,000 μM AEC. Cells were grown at 37°C under 200 rpm in two different regimes: (i) growth for 48 h (single-batch) since the inoculation; (ii) growth for 5 days, with passages to new media every 24 h (100 μl was transferred in each passage). Additionally, wild-type *E. coli* MG1655 cells were also grown for 48 h in minimal media without any AEC present (parent strain genome). Next, the final cultures were streaked to agar plates of the same selective media and single colonies were processed for whole genome sequencing. To do so, genomic DNA was extracted using the Wizard Genomic DNA Purification Kit (Promega), libraries were prepared using the Nextera XT DNA Library Prep Kit (Illumina) and sequenced on an Illumina MiSeq 2x150 paired-end reads run.

Reads were then mapped to the reference *Escherichia coli* str. K-12 substr. MG1655 genome (RefSeq NC_000913.3), using Bowtie2 (v2.3.2) in the *sensitive* preset and *end-to-end* mode. After mapping, SNP calling was done through SAMtools (v1.5) with the following filtering parameters: (i) Phred quality score higher than 20, (ii) SNP read depth higher than 10, and (iii) SNP frequency higher than 50%. Finally, the SNPs called in the sequenced parent genome were subtracted from the SNPs called in the adapted strains, yielding the final list of SNPs (Dataset EV3). The number of reads obtained at each processing step is outlined in Dataset EV4.

## Figure generation softwares

Figures in this work were generated using the matplotlib Python plotting library package (v1.5.3) and Adobe Illustrator CC 2017. Circos plots were generated using Circos (v0.69-3) (Krzywinski *et al*, 2009). Figures with protein structures were generated in PyMOL (v.1.8.6.2).

**Expanded View** for this article is available online.

## Acknowledgements

We thank the University of Colorado Boulder's BioFrontiers Next-Gen Sequencing core facility for assistance and support with the Illumina sequencing runs performed in this study. We also thank the University of Colorado Denver School of Medicine's Mass Spectrometry Facility for assistance with the lysine measurements in this work. This work was supported by the US Department of Energy (Grant DE-SC0008812 and DE-SC0018368), CAPES Foundation (Grant #0315133), and Inscripta, Inc.

## Author contributions

MCB, ADG, and RTG conceived the idea. MCB, AC, WCG, and RTG designed the experiments. MCB and EJO performed the experiments. MCB and AC analyzed the data. WCG contributed to intellectual input, experimental design, and data analyses. ES and TL provided materials and resources. MCB prepared and generated figures. MCB and RTG wrote the manuscript.

## Conflict of interest

The authors declare competing financial interest. The authors R.T.G., A.D.G., E.S., and T.L. have financial interest in the company Inscripta, Inc., which is commercializing the CREATE technology used in this manuscript.

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
