## [Review Process File · Molecular Systems Biology]

Deep scanning lysine metabolism in *Escherichia coli*

Marcelo C. Bassalo, Andrew D. Garst, Alaksh Choudhury, William C. Grau, Eun J. Oh, Eileen Spindler, Tanya Lipscomb and Ryan T. Gill.

Review timeline:	Submission date:	10 th April 2018
	Editorial Decision:	2 nd May 2018
	Revision received:	10 th October 2018
	Editorial Decision:	22 nd October 2018
	Revision received:	26 th October 2018
	Accepted:	30 th October 2018

Editor: Maria Polychronidou

Transaction Report:

1st Editorial Decision

2nd May 2018

Thank you again for submitting your work to Molecular Systems Biology. We have now heard back from the two referees who agreed to evaluate your study. As you will see below, the reviewers acknowledge that the study seems potentially interesting. They raise however a series of concerns, which we would ask you to address in a revision of the manuscript.

Reviewer #2 provides constructive suggestions on additional analyses that will enhance the impact of the study. I think that overall the recommendations of the reviewers are rather clear, but please let me know in case you would like to discuss further any of the reviewers' comments.

REFeree REPORTS

Reviewer #1:

In this work the authors present the application of deep scanning mutagenesis to multiple proteins involved in lysine metabolism with the goal of better understanding the multiple factors that lead to increased production. The work appears to be scientifically sound, and in fact, most of the findings are backed up by other historic studies that have looked at various aspects of this system. Overall, I found the work interesting and clearly presented, but throughout struggled to see the leap in understanding that the paper seemed to suggest in the introduction. I also felt that a more balanced presentation of the methodology in this setting was required to explain not only the potential of the approach, but also the constraints, limits and scalability (see comment 2 below). In light of this, I'd also recommend the authors think about whether the paper might work better if presented more like a method than an article. MSB do have a new category of paper for this.

Below are some major points for consideration:

1. The introduction nicely outlined the importance of considering multiple points in a pathway or complex regulatory network to assess the role of mutations on fitness. Based on this I was expecting to see combinations of mutations explored to look at synergetic or antagonistic effects of changes to multiple elements in the system. However, if I have interpreted the approach correctly, each mutation only affects one gene. While I agree that it's very useful to be able to use pooled sequencing, surely this approach is no different to producing 17 CREATE libraries one for each gene in isolation and then pooling to sequence everything together? Is this approach not significantly hampered by not considering mutations in multiple genes at once? (although this will cause further problems with scalability)
2. One of the challenges with combinatorial approaches is the large numbers of possibilities that arise as the system of interest grows even moderately. As the approach presented here is all about scaling beyond single proteins, how large a system could be investigated with current chip oligo synthesis technologies and sequencing methods? How well does it scale with the number of proteins in a pathway? I would expect to see some calculations and discussion on these properties and specifically the limits of what is currently possible. The paper currently lacks a balanced perspective on limits and capabilities of the approach.
3. When presenting the enrichment of synonymous mutations in LysP you mention some of the mechanisms that might play a role, but do not provide any information as to whether this is supported by, for example, folding energy of the modified transcript, whether the structure of the LysP protein is likely to require co-translational folding (Zhang et al. *Nature Structural and Molecular Biology* 16:274, 2009), or if secondary structures in the transcript might be disrupted and cause issues with translational dynamics (Gorochowski et al. *Nucleic Acids Research* 43:3022-3032, 2015). Some expansion of this analysis would help strengthen the claims made and hopefully better pinpoint the beneficial effect.
4. There was a lack of data regarding the sequencing performed. For example, the number of reads achieved for each replicate, number of mapped reads, number of reads after filtering, etc. This information would be useful to include as a Supplementary File to assess this crucial part of the process.

I also had a few minor comments:

Line 521: the "-16" should be superscript. I'd advise the authors also checking throughout to ensure that subscripts and superscripts are as they should be.

Figure 1D: I didn't find the CREATE workflow particularly clear. It would help to include more details about the steps you take and describing these either visually in the figure or briefly in the caption?

All figures: I found much of the text in the figures barely readable, which makes them frustrating to understand. It would help if it was ensured all text was larger than 6 pt when composed into a figure.

Reviewer #2:

Bassalo et al., manuscript named "Deep scanning lysine metabolism in *Escherichia coli*" is well written and provides new insights in lysine metabolism. Although, this study is a continuation of already published CREATE method, it provides broader insights in one of the applications of the method. Since, the method itself is published, more in depth analysis of proposed application needs to be considered to retain novelty. Please see below a few comments, which in reviewers opinion would strengthen the manuscript.

Major points:

1. Can authors consider combining uncovered mutants in a single strain? Would such a strain have improved production of lysine?
2. L120: To better characterize the method, can authors show if all the residues were targeted in chosen catalytic domains without selection pressure. In other words, did genome edits matched with

the mutagenesis library? If not all the residues were targeted can authors determine the efficiency of mutagenesis.

3. Only a fraction of targeted genes were investigated closer. What was the reason for targeting other genes if they were not investigated after?

4. L290: The benchmarking of ALE with the CREATE method is irrelevant, as all the 19 genes/815 sites selected as CREATE targets are rationally inferred as related to lysine metabolism (L107). This is not the case for ALE, in which both neutral, but also beneficial variants based on seemingly unrelated pleiotropic effects on lysine metabolism may arise. When Figure 6D lists mutations based on 30 hrs this does not take into considerations the 100s of man-years spent investigating the lysine metabolism, which also founded the basis for selection of the 19 genes targeted by CREATE in the first place. The reviewer acknowledges the authors' interest to benchmark the two methods, but the benchmark really does not make much sense beyond serving to illustrate that CREATE allows scanning lysine metabolism "hotspots" in greater depth compared to ALE. Moreover, if including this data in the final manuscript, this reviewer also suggests the authors to put in the lysine quantifications for the 15 whole-genome sequenced strains from the ALE experiment. This indeed would be a relevant benchmark of the two methods.

5. L519: "The intracellular concentration was calculated using the total CFUs present in the pellet (estimated by plating at the harvested stage) and the estimated volume of a single E. coli cell (4.96×10^{-16} L) (Neidhardt & Curtiss, 1996)". Were there no changes in colony sizes observed for the lysine metabolism mutants? If so, the lysine quantifications should be performed relative to dry cell weight and not CFU. Please comment.

Minor points:

1. The font in the abstract does not match.
2. Page 2 lane 51: typo in though.

Reviewer #1:

In this work the authors present the application of deep scanning mutagenesis to multiple proteins involved in lysine metabolism with the goal of better understanding the multiple factors that lead to increased production. The work appears to be scientifically sound, and in fact, most of the findings are backed up by other historic studies that have looked at various aspects of this system. Overall, I found the work interesting and clearly presented, but throughout struggled to see the leap in understanding that the paper seemed to suggest in the introduction. I also felt that a more balanced presentation of the methodology in this setting was required to explain not only the potential of the approach, but also the constraints, limits and scalability (see comment 2 below). In light of this, I'd also recommend the authors think about whether the paper might work better if presented more like a method than an article. MSB do have a new category of paper for this.

Below are some major points for consideration:

1. The introduction nicely outlined the importance of considering multiple points in a pathway or complex regulatory network to assess the role of mutations on fitness. Based on this I was expecting to see combinations of mutations explored to look at synergetic or antagonistic effects of changes to multiple elements in the system. However, if I have interpreted the approach correctly, each mutation only affects one gene. While I agree that it's very useful to be able to use pooled sequencing, surely this approach is no different to producing 17 CREATE libraries one for each gene in isolation and then pooling to sequence everything together? Is this approach not significantly hampered by not considering mutations in multiple genes at once? (although this will cause further problems with scalability)

The reviewer is correct in the statement that combinatorial editing is a powerful approach to investigate and engineer complex phenotypes. While we believe this is a promising future direction of this technology, right now the scalability and editing efficiency are limiting factors as mentioned by the reviewer. To clarify, we included a table (Table 1) in the manuscript providing more detailed metrics on an estimated coverage and editing efficiency in our studies. We note that introducing single amino

acid changes across the genome is particularly valuable as this allows the investigation of the contribution of individual mutations to the phenotype of interest, which is often a starting point in more sophisticated combinatorial search strategies.

2. One of the challenges with combinatorial approaches is the large numbers of possibilities that arise as the system of interest grows even moderately. As the approach presented here is all about scaling beyond single proteins, how large a system could be investigated with current chip oligo synthesis technologies and sequencing methods? How well does it scale with the number of proteins in a pathway? I would expect to see some calculations and discussion on these properties and specifically the limits of what is currently possible. The paper currently lacks a balanced perspective on limits and capabilities of the approach.

As mentioned in comment #1, we included a table (Table 1) containing more detailed metrics on actual genomic edits that we observed in the current study. We also included extensive changes in the manuscript (L291-336; L391-423) with a more thorough discussion on the current limits and capabilities of this approach. In these sections, we discuss some of the concerns raised by the reviewer such as sequencing depth and scalability. Further, we suggest several technical and experimental changes that could alleviate some of the current limitations in future implementation of this technology.

3. When presenting the enrichment of synonymous mutations in LysP you mention some of the mechanisms that might play a role, but do not provide any information as to whether this is supported by, for example, folding energy of the modified transcript, whether the structure of the LysP protein is likely to require co-translational folding (Zhang et al. Nature Structural and Molecular Biology 16:274, 2009), or if secondary structures in the transcript might be disrupted and cause issues with translational dynamics (Gorochowski et al. Nucleic Acids Research 43:3022-3032, 2015). Some expansion of this analysis would help strengthen the claims made and hopefully better pinpoint the beneficial effect.

This is a great suggestion by the reviewer. An expansion on this topic was added to the manuscript (L225-230).

4. There was a lack of data regarding the sequencing performed. For example, the number of reads achieved for each replicate, number of mapped reads, number of reads after filtering, etc. This information would be useful to include as a Supplementary File to assess this crucial part of the process.

We included a new table (Dataset EV4) with descriptive information on our sequencing runs.

I also had a few minor comments:

Line 521: the "-16" should be superscript. I'd advise the authors also checking throughout to ensure that subscripts and superscripts are as they should be.

We fixed these issues throughout the manuscript.

Figure 1D: I didn't find the CREATE workflow particularly clear. It would help to include more details about the steps you take and describing these either visually in the figure or briefly in the caption?

A more detailed description on the actual steps was included in the figure caption.

All figures: I found much of the text in the figures barely readable, which makes them frustrating to understand. It would help if it was ensured all text was larger than 6 pt when composed into a figure.

Font size for all figures was increased.

Reviewer #2:

Bassalo et al., manuscript named "Deep scanning lysine metabolism in Escherichia coli" is well written and provides new insights in lysine metabolism. Although, this study is a continuation of already published CREATE method, it provides broader insights in one of the applications of the method. Since, the method itself is published, more in depth analysis of proposed application needs to be considered to retain novelty. Please see below a few comments, which in reviewers opinion would strengthen the manuscript.

Major points:

1. Can authors consider combining uncovered mutants in a single strain? Would such a strain have improved production of lysine?

As suggested, we performed new experiments to combine several of the mutants described in this manuscript into a single strain. We did not observe any further improvements in the combined strains. This is not surprising given the complex set of parameters that govern pathway flux (we observed similar neutral or even antagonistic effects in our previous studies along these lines, Sandoval et al., PNAS, 2012). We agree that a combinatorial engineering approach would be extremely valuable and is a promising future direction from technologies such as CREATE. Such an approach requires, however, that search strategies are employed that attempt to explicitly consider pleiotropic effects rather than the simple combining of the best individual mutants (as we show here, and in more depth in our prior PNAS paper cite above).

2. L120: To better characterize the method, can authors show if all the residues were targeted in chosen catalytic domains without selection pressure. In other words, did genome edits matched with the mutagenesis library? If not all the residues were targeted can authors determine the efficiency of mutagenesis.

This is a great suggestion from the reviewer, which we addressed with new experimentation. Although a full assessment of the genomic edits and library coverage would not be feasible on a genome-wide scale (hence the value of technologies such as CREATE), we selected a few regions from targeted genes to deep sequence and investigate actual genomic edits. The results from this analysis was included in Table 1 and as Figure EV2 in the manuscript.

3. Only a fraction of targeted genes were investigated closer. What was the reason for targeting other genes if they were not investigated after?

We wanted to target all genes related to lysine metabolism so that they could compete against each other under selection. That way, we could assess the relative contribution of each gene to the selective pressure. We could not investigate all targeted genes in more details for two main reasons. First, validation and mechanistic elucidation of target edits is the most time-consuming step, with the study of many of these genes in isolation being a manuscript itself. Therefore, we believe that mapping of multiple of these genes in parallel followed by validation of a few of them, picked to highlight different aspects of the technology, is in itself a significant contribution. Second, current limits on depth and noise prohibits detailed investigation of every single designed edit. We included an extensive discussion on this issue to clarify the current limitations and provide a more balanced manuscript.

4. L290: The benchmarking of ALE with the CREATE method is irrelevant, as all the 19 genes/815 sites selected as CREATE targets are rationally inferred as related to lysine metabolism (L107). This is not the case for ALE, in which both neutral, but also beneficial variants based on seemingly unrelated pleiotropic effects on lysine metabolism may arise. When Figure 6D lists mutations based on 30 hrs this does not take into considerations the 100s of man-years spent investigating the lysine metabolism, which also founded the basis for selection of the 19 genes targeted by CREATE in the first place. The reviewer acknowledges the authors' interest to benchmark the two methods, but the benchmark really does not make much sense beyond serving to illustrate that CREATE allows scanning lysine metabolism "hotspots" in greater depth compared to ALE. Moreover, if including this data in the final manuscript, this reviewer also suggests the authors to put in the lysine quantifications for the 15 whole-genome sequenced strains from the ALE experiment. This indeed would be a relevant benchmark of the two methods.

We agree with the reviewer that CREATE and ALE are different strategies, and our intentions by comparing them was not to suggest CREATE as a replacement (or better) approach than ALE. The value of ALE is beyond discussion, with decades of contribution to the field. As the reviewer mentioned, CREATE allowed us to scan with

greater depth “pre-selected hotspots”. We do not believe CREATE and ALE are mutually exclusive approaches, and in fact a combination of both would be extremely valuable. We included changes in this section (L357-362) to better highlight this in the manuscript.

5. L519: "The intracellular concentration was calculated using the total CFUs present in the pellet (estimated by plating at the harvested stage) and the estimated volume of a single E. coli cell (4.96×10^{-16} L) (Neidhardt & Curtiss, 1996)". Were there no changes in colony sizes observed for the lysine metabolism mutants? If so, the lysine quantifications should be performed relative to dry cell weight and not CFU. Please comment.

Intracellular lysine was measured from pelleted cells of a liquid culture, with the number of cells normalized by OD. We plated a fraction of the cultures just to estimate the number of cells per mL in the culture, in order to allow the calculation of absolute values. That way, the size of the colonies should not interfere, as this is a consequence of plate incubation time/cell growth rate, while lysine was extracted from cultures before plating, which were all normalized to the same OD. Also, if the number of CFUs is off, the absolute values would change, but the relative amounts between samples would not. Considering that the absolute values are based on estimated parameters that we did not measure (such as cell volume), we decided to change this in the manuscript and report lysine as fold-change to wild-type levels.

Minor points:

1. The font in the abstract does not match.

We fixed this in the revised manuscript.

2. Page 2 line 51: typo in though.

Typo was fixed in the revised manuscript.

Thank you again for sending us your revised study. We have now heard back from reviewer #2 who was asked to evaluate your study. As you will see below, the reviewer thinks that all issues have been satisfactorily addressed and is supportive of publication.

Before we formally accept your manuscript for publication, we would ask you to address a couple of remaining editorial issues.

REFEREE REPORTS

Reviewer #2:

All the questions previously raised have been sufficiently addressed.

YOU MUST COMPLETE ALL CELLS WITH A PINK BACKGROUND ↓
PLEASE NOTE THAT THIS CHECKLIST WILL BE PUBLISHED ALONGSIDE YOUR PAPER

Corresponding Author Name: Ryan T. Gill
Journal Submitted to: Molecular Systems Biology
Manuscript Number: MSB-18-8371